

# TrackMatcher — A tool for finding intercepts in tracks of geographical positions

Peter Bräuer[1] and Matthias Tesche[1]

[1]Leipzig University, Institute for Meteorology, Stephanstraße 3, 04103 Leipzig, Germany

**Correspondence:** Matthias Tesche (matthias.tesche@uni-leipzig.de)

**Abstract.** Working with measurement data in atmospheric science often necessitates the collocation of observations from instruments or platforms at different locations, with different geographical and/or temporal data coverage. Varying complexity and abundance of the different data sets demand a consolidation of the observations. This paper presents a tool for (i) finding temporally and spatially resolved intersections between two- or three-dimensional geographical tracks (trajectories) and (ii) extracting of observations and other derived parameters in the vicinity of intersections to achieve the optimal combination of various data sets and measurement techniques.

The TrackMatcher tool has been designed specifically for matching height-resolved remote-sensing observations along the ground track of a satellite with position data of aircraft (flight tracks) and clouds (cloud tracks) and intended extension for ships (ship tracks) and air parcels (forward and backward trajectories). The open-source algorithm is written in the Julia programming language. The core of the matching algorithm consist of interpolating tracks of different objects with a piecewise cubic Hermite interpolating polynomial with subsequent identification of an intercept point by minimising the norm between the different track point coordinate pairs. The functionality wrapped around the two steps allows for application of the TrackMatcher tool to a wide range of scenarios. Here, we present three examples of matching satellite tracks with the position of individual aircraft and clouds that demonstrate the usefulness of TrackMatcher for application in atmospheric science.

## 1 Introduction

In atmospheric science, data from different measurement platforms or locations are often combined for synergistic analysis or validation purposes. This holds particularly for the combination of measurements from different spaceborne sensors, e.g., Sun-Mack et al. (2007); Kato et al. (2011); Redemann et al. (2012) or Alfaro-Contreras et al. (2017), and the long-term validation of those measurements at ground sites, e.g. Pappalardo et al. (2010); Tesche et al. (2013). The collocation problem is particularly relevant for mobile observations (airborne, ship-based or space-borne) with active sensors such as lidar or radar. In contrast to passive sensors with a swath width on the order of 1000 km, active sensors provide height-resolved measurements along a very narrow ground-track, so-called curtain observations below or above the track of the platform that carries the respective instrument.

In the past, observations with the Cloud-Aerosol Lidar with Orthogonal Polarization (CALIOP) aboard the Cloud-Aerosol Lidar and Infrared Pathfinder Satellite Observations (CALIPSO; Winker et al., 2009) satellite or the Cloud-Profiling Radar



(CPR) aboard the CloudSat satellite (Stephens et al., 2002) have been matched with (i) backward (forward) trajectories arriving (starting) at a specific ground site (Tesche et al., 2013, 2014), (ii) ship tracks in decks of stratiform clouds (Christensen and Stephens, 2011), (iii) linear contrails as identified from passive remote sensing (Iwabuchi et al., 2012) or (iv) flight tracks of aircraft (Tesche et al., 2016). The TrackMatcher tool has been developed to enable a unified and objective way of finding

temporal and spatial matches between the ground-tracks of satellites or research aircraft that perform height-resolved observations of atmospheric parameters and spatio-temporal information (tracks) of ships, aircraft, clouds, or air parcels. In addition to the information on time and location that is needed to perform the matching, the algorithm handles auxiliary data along the considered tracks and enables collocation and sub-sampling of the along-track data sets. The technical details and the performance of the TrackMatcher algorithm are described in Sect. 2. Examples of the application of the tool are provided in Sect. 3.

We conclude this work with a summary and an outlook of potential fields of application of the developed algorithm in Sect. 4.

## 2 The TrackMatcher package

### 2.1 Motivation

The purpose of TrackMatcher is the identification of intercept points between two time-resolved three-dimensional paths of latitude/longitude/height ($\varphi/\lambda/h$) coordinates (referred to as tracks or trajectories from now) and the collection of the

respective data fields along those tracks. Tracks may be reduced to two dimensions, e.g. for objects moving at ground level or, in the case of satellite data, saved as "curtains", where the column profile above the ground track position is stored.

TrackMatcher operates on a *primary* data set with individual trajectories and matches them to a continuous trajectory in a *secondary* data set. For performance reasons, the secondary trajectory is stored as segments, which enables the TrackMatcher package to be easiliy expanded to compare to two sets of individual trajectories. A detailed overview of the data structure

in TrackMatcher is given in Sect. 2.3. The package features options regarding (i) the format of the input data from text files with comma- (csv) or tab-separated values (tsv) or from HDF4 or MATLAB's *mat* files, (ii) the configuration of the output fields, and (iii) the optimisation of the balance between performance of the algorithm and accuracy of the results. A detailed description of the settings can be found in Sect. 2.5.

While we refer to the general terms of primary and secondary data in this paper, the motivation for developing TrackMatcher

was the desire to find intersections between the three-dimensional flight tracks of individual aircraft (primary data) and curtain observations along satellite ground-tracks (secondary data). Specifically, the position of aircraft and auxiliary information such as the type of aircraft, engine, and fuel should be matched to vertically resolved extinction coefficients from a spaceborne lidar measurements to assess the environmental and climate impact of an aircraft passage through a cirrus cloud (Tesche et al., 2016).

For this purpose, TrackMatcher ought to process two types of spaceborne observations along the satellite ground track related to information on (i) column parameters for cloud or aerosol layers and (ii) vertically resolved observations (profiles) within those cloud or aerosol layers. The algorithm was designed to operate with aircraft location data as available from online flight trackers (e.g. https://flightaware.com/) or databases that provide position data for individual aircraft (Brasseur et al., 2016). The





large volume of the considered position data for matching with an abundance of satellite track data together with the overall
low match rate of the two requires an automated and objective procedure that is realised in the TrackMatcher tool.

Despite the initially highly specific scope for developing TrackMatcher, the tool is useful for a much wider range of applications that require matching time, position, and auxiliary data along two tracks on a geographic grid. Potential applications include matching vertically resolved information along satellite tracks with (i) tracks of individual clouds (see Sect. 3.3 and Seelig et al., 2021), (ii) ship tracks, or (iii) trajectories of air parcels from dispersion modelling; or even (iv) matching three-
dimensional flight data with three-dimensional cloud tracks. While the current focus of TrackMatcher is on applications in atmospheric science, the tool is designed for great flexibility with respect to the input data for further applications in the wider geosciences.

## 2.2   Code availability and package dependencies

TrackMatcher is an open source package hosted at GitHub (https://github.com/LIM-AeroCloud/TrackMatcher.jl.git) under the
GNU General Public Licence v3.0. We strongly encourage contributions from outsiders, e.g., by pull requests or filing issues.

TrackMatcher is written in Julia (Bezanson et al., 2017). The package relies on MATLAB to read the satellite data from HDF4 files. The software is distributed as an unregistered Julia package and is tested against Julia 1.6.3 and the most recent stable release (currently identical version). Besides a Julia and MATLAB installation, the following Julia package dependencies exist with the version numbers given in parentheses:

– MATLAB (v0.8.2)

– MAT (v0.10.1)

– CSV (v0.9.10)

– DataFrames (v1.2.2)

– DataStructures (v0.18.10)

– IntervalArithmetic (v0.20.0)

– IntervalRootFinding (v0.5.10)

– Distances (v0.10.5)

– TimeZones (v1.6.2)

– PCHIP (v0.2.1)

– ProgressMeter (v1.7.1)

The following modules are used from Julia's base:





- Dates

- Logging

- Statistics

The PCHIP package was developed within the TrackMatcher framework to allow track interpolation with a piecewise cubic Hermite interpolating polynomial (see Sect. 2.4). It is available under the GNU General Public Licence version 3 at Github (https://github.com/LIM-AeroCloud/PCHIP.jl.git).

## 2.3   Data structure

TrackMatcher is organised in data sets making use of Julia's type system. For readers unfamiliar with the type system, Sect. S1.1
in the ESM highlights the key points of this ecosystem and the different types.

Figure 1 shows TrackMatcher's type tree. Green boxes in Fig. 1 are used for concrete types that store track data or observations. Dark Blue boxes denote abstract types needed to classify the concrete types. Light blue boxes show an abstract type (top white label) together with an concrete type (bottom grey label). These are special cases for abstract types within the type tree that hold a concrete type as child. Both, abstract and concrete type have a constructor to instantiate the data in the concrete
type tying both types together. For boxes circled in violet, a convenience constructor exists that initiates a TrackMatcher process such as loading input data and/or calculating intersections. Section S1.2 in the ESM presents details of the data structure introducing field names and data types for each struct.

In TrackMatcher, the top level type is a generic *DataSet*, which holds *Data* with fields for measured (*MeasuredSet*) and computed (*ComputedSet*) data. The *MeasuredSet* organises the track data and observations. Observations are subtypes of
*ObservationSet* as shown in Fig. 1. Track data are split into primary and secondary data sets. Currently, primary data consist of individual trajectories (a *PrimaryTrack*) that are combined in a *PrimarySet*. The current version of TrackMatcher distinguishes between flight and cloud data in the *PrimarySet* and *PrimaryTrack*. Within the *FlightSet*, a *FlightTrack* can be obtained from several sources, however, the format of *FlightData* is unified. Currently, the only *SecondarySet* is *SatSet*. It holds data from a continuous trajectory, which is split into segments. Segments are stored in the field granules of *SatSet*. Segments or granules
are classified below the *SecondarySet* level as *SecondaryTrack*, currently only holding *SatData* of the *SatTrack*. Each *SatTrack* is a track segment of the satellite track from either the day- or night-time hemisphere of Earth loaded from the individual input files. *Intersection* data (or *XData*) with intercept points from the primary and secondary trajectories are stored as child of *ComputedSet* with no distinction which primary data type was used for the calculation.

## 2.4   Algorithm description

The key steps of the algorithm are to (i) load track data related to two platforms, (ii) interpolate the individual tracks, (iii) find intersections by minimising the norm between the different track point coordinate pairs, and (iv) extracting auxiliary information at or around the intercept point as set by the operator. These steps are described in the subsequent subsections with examples for matching aircraft or cloud position data with satellite ground tracks given in Sect. 3 as motivated by Sect. 2.1.





**Figure 1.** Type tree used in the TrackMatcher package.

Intersections between two tracks are defined as those locations for which the distance between a pair of points from the primary and secondary track reaches zero. This distance can be calculated either as the difference of the latitude values from two tracks with a common longitude value or as the difference of the longitude values for a common latitude value. The general





form of the distance function is

$$d(x) = \tau_{\mathrm{prim}}(x) - \tau_{\mathrm{sec}}(x). \tag{1}$$

Here, $x$ is either a latitude or longitude value and $\tau_{\mathrm{prim/sec}}$ represent the primary and secondary trajectories that determine the
corresponding longitude and latitude values. The roots of Equation 1 define intercept points between both tracks.

### 2.4.1 Track data import

Primary and secondary data are loaded from provided files (currently csv, tsv, mat or hdf) and stored in a unified format in
the respective structs of the type tree presented in Fig. 1. Time-resolved track points of the individual trajectories and other
relevant data at these points are stored in a field *data* that consists of a DataFrame with columns for each property and rows for
every time step. Additional database information and overall information concerning the trajectory as a whole are stored in the
metadata field of the primary data.

For the primary data, individual tracks from one source are loaded to a vector, which in turn are combined in the PrimarySet
of the respective data type (FlightSet or CloudSet). Each PrimarySet can have several fields for vectors of PrimaryTrack. The
structures of the primary data currently available in TrackMatcher according to the examples presented in Sect. 3 are visualised
in the ESM in Fig. S3.

Secondary track data are stored as segments of a continuous track for performance reasons (see also Sect. 2.3). All track
segments are combined in a set. Secondary track data (currently only *SatData*) are stored in a *data* field with a DataFrame using
the same structure as primary data (see Fig. S2 for schematics of the current data structure). Only essential data needed for the
calculation of intersections are stored in this struct for performance reasons, i.e. time, latitude, and longitude. All SatData are
combined in the field *granules* of the *SatSet*. Additional information about the temporal and spatial coverage of each granule
and the location of the input file is given in the SatSet metadata. Observations at track points are only extracted from the input
files, if intercept points between the primary and secondary track sets are found. Specifics regarding the extracted data can be
customised for the desired application.

### 2.4.2 Track data interpolation

A fundamental problem of the TrackMatcher algorithm is that the true functions of the investigated tracks are unknown and
have to be approximated. The approximation is only valid near nodes, i.e. in the vicinity of known track points. For Eq. 11 to be
applicable, $x$ (either latitude or longitude) must be equal for both tracks. In reality, the two tracks rarely feature regular intervals
and most likely will not share a common latitude or longitude vector. Hence, both data sets will first need to be interpolated
with a common set of $x$ data between a shared start and end value.

Generally, track data cannot be fitted to a known function and the connection between latitude/longitude pairs needs to be
approximated with a suitable interpolation method. We chose the Piecewise Cubic Hermite Interpolating Polynomial (PCHIP;
see, e.g., Fritsch and Carlson, 1980; Kahaner et al., 1989; Turley, 2018) to approximate a function $f(x)$ between any two data
points $x_i$ and $x_{i+1}$ as





$$f(x) = f(x_i)H_1(x) + f(x_{i+1})H_2(x) + f'(x_i)H_3(x) + f'(x_{i+1})H_4(x). \tag{2}$$

Polynomials in Eq. (2) are defined as

$$H_1(x) = \phi\left(\frac{x_{i+1} - x}{h_i}\right) \tag{3}$$

$$H_2(x) = \phi\left(\frac{x - x_i}{h_i}\right) \tag{4}$$

$$H_3(x) = -h_i\psi\left(\frac{x_{i+1} - x}{h_i}\right) \tag{5}$$

$$H_4(x) = h_i\psi\left(\frac{x - x_i}{h_i}\right) \tag{6}$$

and

$$h_i = x_{i+1} - x_i \tag{7}$$

$$\phi(x) = 3x^2 - 2x^3 \tag{8}$$

$$\psi(x) = x^3 - x^2. \tag{9}$$

The PCHIP method demands a continuous first derivative $f'(x)$ at each data point (node), but in contrast to cubic splines does not require a continuous second derivative $f''(x)$. This principally means that cubic splines are slightly more accurate in approximating continuous curves. For our purpose, the decreased accuracy is negligible and well within the errors of the track data points themselves. Instead, PCHIP interpolation suppresses artificial oscillation at discontinuities, which can occur at sharp turns of a track.

Track interpolation is a comprehensive task that consumes a significant portion of the source code. Therefore, PCHIP has been outsourced as a separate package available at GitHub under the GNU general public licence version 3.0 or above (https://github.com/LIM-AeroCloud/PCHIP.jl.git).

### 2.4.3 Calculation of intersept points

To find intercept points between the primary and secondary tracks, the TrackMatcher algorithm follows seven steps that are explained in more detail below:

1. Load primary data, find the prevailing track direction and inflection points in the $x$-data of the primary track.

2. Load secondary data for the time frame of the primary data including a pre-defined tolerance at the beginning and end; find inflection points in the $x$-data based on the prevailing track direction of the primary data.

3. Put a bounding box around the coordinates of the primary track (see Fig. 2) and find segments of the secondary track within these coordinates and a given window of acceptalbe temporal difference $\pm\Delta t$.





4. Interpolate the track segments of the primary and secondary tracks with the PCHIP method using common equidistant $x$-data with a defined step width.

5. Define a function (Eq. (1)) to obtain roots between the difference in the track points of both tracks.

6. Roots of Eq. (1) are intersections between both tracks.

7. Filter intersections. Save intersection data and relevant measurements from the input data in the vicinity of the intercepts.

The result of the procedure is visualised in an example of an aircraft flight track and a satellite ground track in Fig. 2.

Track data are loaded (*steps 1 and 2*) as explained in Sect. 2.4.1 and 2.5.2. For the interpolation of the track data of both trajectories in step 4, strictly monotonic ascending $x$-data are a requirement. Therefore, both trajectories are fragmented into segments that fulfil this condition. To minimise fragmentation, the prevailing direction (north—south or east—west) of the primary track is determined while reading the input data according to

$$\delta(\varphi) \leq (\delta(\lambda^+) + \delta(\lambda^-)) \cdot \cos(\overline{\varphi}) \tag{10}$$

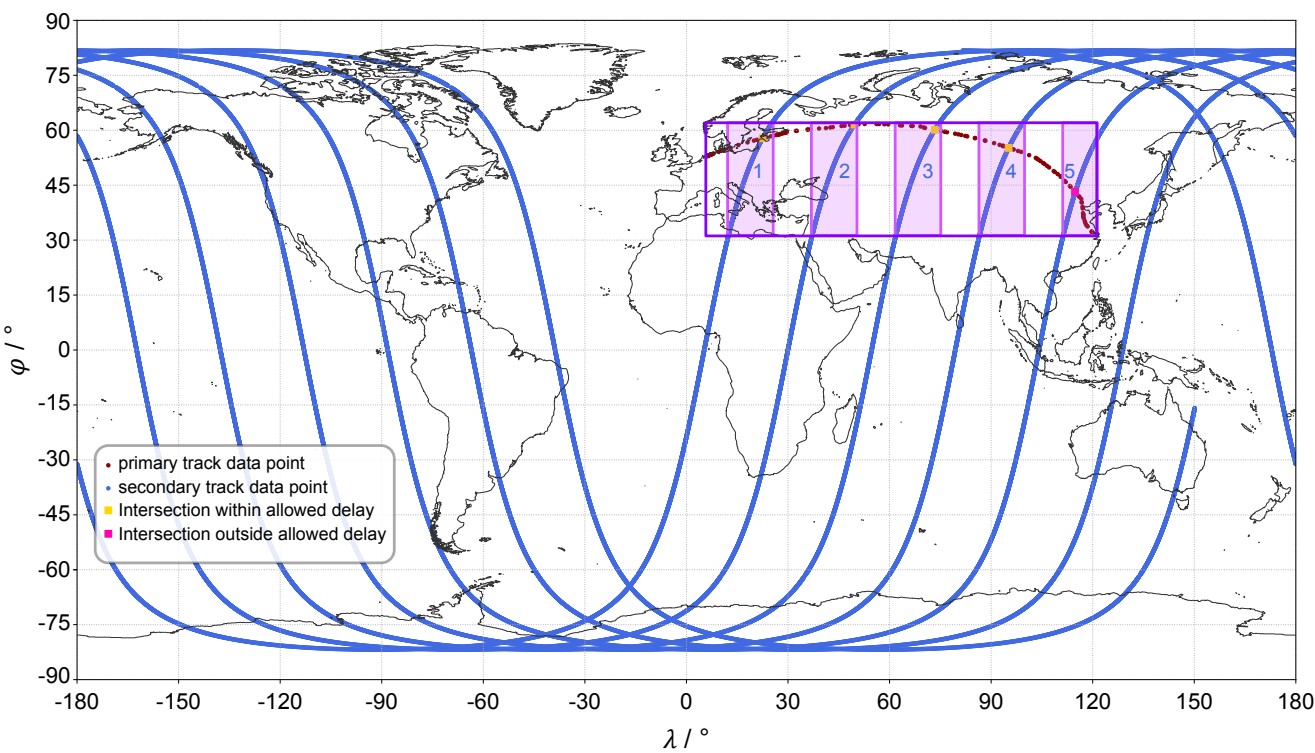

**Figure 2.** Example plot of flight (primary) and satellite (secondary) track data and the intersections found by the algorithm.





with

$$\delta(x) = \max(x) - \min(x). \tag{11}$$

Longitude $\lambda$ is chosen as $x$ value, if Eq. (10) is true and the dominant direction of the primary track is east–west. Otherwise, latitude $\varphi$ performs better for a prevailing north–south direction. In Eq. (10), maximum horizontal distances are calculated

separately for positive ($\lambda^+$) and negative ($\lambda^-$) longitude values to avoid problems with the sign change at the date line. The last term on the right-hand side of Eq. (10) corrects for the poleward-decreasing distance between meridians. For simplicity, only the mean latitude ($\overline{\varphi}$) is considered. As highly irregular patterns are currently only expected in the primary track, their data defines, whether $\varphi$ or $\lambda$ is used as $x$-data in both tracks.

Before interpolation, data of the secondary track need to be extracted for the correct time frame and location in *step 3*.

Therefore, only data points are considered that are within the time span of the primary data. Additional data points at the beginning and end are allowed to permit a delay at possible intercept points of the primary and secondary track. By default, a maximum delay of 30 minutes is permitted, which can be adjusted by the user. To exclude unnecessary secondary track points, a bounding box is put around the primary track as shown in Fig. 2. Only track segments of the secondary trajectory within this bounding box are considered in any further steps (visualised by the pink boxes 1 to 5 in Fig. 2). To allow for rounding errors,

the bounding box can be increased by an absolute tolerance (by default $0.1°$).

In *step 4*, all segments are interpolated. Common $x$-data are used in the overlap regions (pink boxes in Fig. 2). These segments ($\tau_{prim}/\tau_{sec}$ in Eq. (1)) are used for the identification of intercept points. Eq. (1) is case-specific and is re-defined for every segment pair (or box) in *step 5*. In *step 6*, TrackMatcher uses the *IntervalRootFinding.jl* package to determine all roots of Eq. (1). These roots are intercept points between the tracks of the primary and secondary data set.

In rare cases, the algorithm duplicates intercept points. This occurs mainly when an intercept point is located at a track segment boundary where matches are then found at either side of the segment border. Duplicate intercepts can also occur in case of near-parallel tracks. To avoid the output of duplicate detection, the algorithm verifies that only the intercept calculation with the highest accuracy is stored within a predefined radius. By default, this radius is $20\,\mathrm{km}$, but can be customised by the user.

Duplicate intercept points as well as intercept points for which the delay between the primary and secondary track exceeds a set time difference are disregarded in *step 7* (pink square in Fig. 2). This and further filtering is controlled by keyword arguments at the start of a TrackMatcher run as outlined in Table 1. For the remaining intercept points (yellow squares in Fig. 2), the latitude, longitude, respective times of the primary and secondary track at the intercept point as well as the difference between these times, details regarding the accuracy of the calculation, and user-selected auxiliary data in the vicinity of the intercept

points are saved.

Linear interpolation between recorded time steps of the track points is used to derive the exact intercept time. Accurate interpolation as with the PCHIP method demands track segments with strictly monotonic latitude and longitude data (see Sect. 2.4.2). This likely results in a strong segmentation of the original track and leads to high computational costs. However, satellites, clouds, and aircraft at cruising altitudes are expected to move with relatively constant velocity, which is why linear





interpolation can be applied to derive the time of intercept. For ascending and descending aircraft with unknown acceleration or deceleration, track points are close to each other leading to minimal errors from linear interpolation.

To calculate the distance between track points, which is needed for setting a variety of thresholds, we use the haversine function

$$d = 2r \arcsin\left(\sqrt{\sin^2\left(\frac{\varphi_2 - \varphi_1}{2}\right) + \cos(\varphi_1)\cos(\varphi_2)\sin^2\left(\frac{\lambda_2 - \lambda_1}{2}\right)}\right), \tag{12}$$

which gives the great circle distance between two points on a perfect sphere. In Eq. (12), $\varphi_{1/2}$ are the latitude values of the coordinate pairs 1 and 2, $\lambda_{1/2}$ are the respective longitude values, and $r$ is the radius of a perfect sphere. The poleward decrease of the Earth's radius is approximated as

$$r(\varphi) = \sqrt{\frac{\left(r_{eq}^2 \cos(\varphi)\right)^2 + \left(r_{pol}^2 \sin(\varphi)\right)^2}{\left(r_{eq} \cos(\varphi)\right)^2 + \left(r_{pol} \sin(\varphi)\right)^2}} \tag{13}$$

with $r_{eq} = 6\,378\,137\,\mathrm{m}$ and $r_{pol} = 6\,356\,752\,\mathrm{m}$ as the radii in the equatorial and polar plane, respectively.

## 2.5 Programme description

### 2.5.1 General information and package installation

This paper describes TrackMatcher version 0.5.3 and PCHIP version 0.2.1. A wiki with a complete manual is available from the TrackMatcher repository at GitHub. Here, only the most important aspects of the tool and its usage are highlighted. The programme description is not meant to be a complete set of instructions. Further guides and examples can be found in Sect. 2 of the ESM.

With version numbers below 1.0, breaking changes may still occur frequently at the introduction of new minor versions. However, this eases the introduction of new features or the improvement of current routines. Contributions from outsiders, e.g., by pull requests or filing issues, are strongly encouraged to enhance the performance and flexibility of the algorithm.

Both, TrackMatcher and PCHIP are unregistered Julia packages. The easiest way to install them is by using Julia's inbuilt package manager and adding the url of the GitHub repository. Example code for TrackMatcher installation is shown in Script 1 in Sect. S2.1 of the ESM. This will install the package and all dependent registered Julia packages in the correct version as defined by the package's project file. However, dependent unregistered packages need to be installed manually prior to the actual package installation. Therefore, the order of installation for TrackMatcher must be:

1. PCHIP

2. TrackMatcher

Package developers can use the *dev* option from Julia's package manager. However, it is recommended to clone the TrackMatcher repository from GitHub and activate the main folder with the Project.toml file to develop and run TrackMatcher.





**Table 1.** Parameters controlling TrackMatcher runs. Arguments are printed italic, keyword arguments use roman font.

| Parameter | Unit | Data type | Default | Meaning |
|---|---|---|---|---|
| **Parameters for *FlightSet, CloudSet, SatSet, and Intersection*** | | | | |
| savedir | — | Union{String,Bool} | "abs" | Save directories and file names as absolute paths (*"abs"*, default), relative paths (*"rel"*) or as given by user without saving additional observations ("" or *false*). |
| remarks | — | Any | nothing | Any remarks or data attached to metadata. |
| **Parameters for *FlightSet*** | | | | |
| altmin | m | Real | 5000 | Threshold for minimum altitude in meters below which track data are ignored. |
| odelim | — | Union{Nothing,Char,String} | nothing | Delimiter for csv input files regarding *webdata* in *FlightSet*. The default value *nothing* causes automatic delimiter detection by the CSV package. |
| **Parameters for *CloudSet*** | | | | |
| structname | — | String | "cloud" | Name for top layer struct in mat files with cloud tracking data. |
| **Parameters for *SatSet*** | | | | |
| type | — | Symbol | undef | Used to enforce loading either cloud layer (CLay) or profile (CPro) data, otherwise the type is inferred from the majority of the first 50 input files. |
| **Parameters for *Intersection*** | | | | |
| *savesecondsattype* | — | Bool | false | When set to true, saves both cloud layer and profile data, by default only the type used in SatSet is saved. |
| maxtimediff | min | Int | 30 | Accepted time delay between the primary and secondary tracks at intercepts. |
| primspan | — | Int | 0 | Stores ±primspan data points along the primary track closest to the intercept point (0 = only the data point closest to the intercept is saved). |
| secspan | — | Int | 15 | Stores ±secspan data points along the primary track closest to the intercept point (0 = only the data point closest to the intercept is saved). |
| lidarrange | m | Tuple{Real,Real} | (15 000,-Inf) | (upper, lower) height level between which CALIPSO data are considered; (Inf, -Inf) considers all data. |
| stepwidth | ° | Real | 0.01 | Step width used in interpolated tracks. |
| Xradius | m | Real | 20000 | Radius around an intercept point within which only the most accurate intercept computation is considered for multiple finds. |
| expdist | m | Real | Inf | Calculations for which the closest measured track point is above the expdist threshold are disregarded (Inf = all calculations are considered). |
| atol | ° | Real | 0.1 | Absolute tolerance to increase the bounding box around primary tracks. |



Moreover, TrackMatcher relies on a Julia installation of at least version 1.6 with long-term support given for version 1.6 and further support for the current stable minor release. Additionally, TrackMatcher requires a licensed MATLAB version. If
your MATLAB version is not installed to the standard directory of your system, it needs to be linked to Julia as described by Julia's MATLAB package README (https://github.com/JuliaInterop/MATLAB.jl.git). Further help for linking MATLAB to Julia can be acquired from the package's resources.

### 2.5.2 Loading input data

Currently, TrackMatcher is configured to process three types of data:

– aircraft track data from different sources (stored as primary data in *FlightData*/*FlightSet*)

     – cloud track data (stored as primary data in *CloudData*/*CloudSet*)

     – CALIPSO satellite data (stored as secondary data in *SatData*/*SatSet*)

Track data are stored in structs with a data field and, for primary data, a metadata field. Metadata is another struct with information about the raw data, computation settings and times, and properties concerning the whole trajectory. Time-resolved
data are stored in a DataFrame in the data field with columns *time*, *lat*, and *lon*, and further columns depending on the data type (aircraft, cloud or satellite data).

Tracks from primary data sets are stored in individual structs, which are combined in a vector and stored in a database struct (FlightSet or CloudSet) together with metadata (see Sect 2.3 and ESM for details). Several databases, e.g. individual flight tracks saved in tsv files or complex flight inventories, are considered for aircraft data. Each database type is stored in a separate
vector/FlightSet field.

In contrast, secondary data consists of a single long trajectory. However, for performance reasons, data are stored as *SatData* structs for individual granules (track segments holding data of either the day- or night-time hemisphere of Earth). Moreover, only time, latitude, and longitude are stored in SatData for an optimised performance.

Further satellite data are only loaded in the vicinity of intersections. Currently, additional satellite data can be extracted either
as a height profile (*CPro*) or as a layer mean value (*CLay*). The additional data are also used to determine the meteorological conditions at the intersection. Only one type of the observation data (CPro or CLay) can be used to derive intercept points. The data type is determined automatically from the keyword *CPro* or *CLay* embedded in the file names. If both types exists, the data type with a majority in the first 50 files is chosen unless the default behaviour is overwritten by user settings. It should be noted that profile data gives a more refined height resolution and, hence, a more precise representation of the height-resolved
atmospheric state at the intersection at the cost of more memory usage and a longer computation time.

Data are loaded from HDF4, MATLAB data (.mat) files or text files with comma-separated values (csv) or tab-separated values using "tsv", "dat" or "txt" as file extension. Details on the database types, file formats, restrictions and conventions can be found in Sect. S2.2.1 of the ESM.





To load data into the respective struct a convenience constructor exists that takes any number of file strings with absolute
or relative folder paths as input. These directories and all subfolders are searched recursively for any file with the correct
extension. These files are assumed to be valid data files or will produce a non-critical error during loading. Further keyword
arguments control data reading and filtering as given in Table 1.

Aircraft data are currently the only data type with multiple database sources. For the data files, csv is commonly used as
file format. Therefore, data files cannot be identified by file extension alone as this does not allow an assignment to the correct
database. Therefore, directories are passed as strings to keyword arguments for the respective database type. If a user wants
to scan more than one directory for the same database type, a vector of strings can be passed to the keyword arguments (see
Sect. S2.2 and Note 7 in the ESM for details).

### 2.5.3 Calculating intersections and model output

To calculate intersections between the trajectories of the primary and secondary data sets, the user only needs to instantiate
a new *Intersection* struct using either FlightSet or CloudSet and SatSet as input to a modified constructor for Intersection.
The algorithm works only for either flight or cloud data and two intersection structs need to be instantiated, if you want to
calculate intersections for both types. However, the algorithm does not differentiate between the different flight database types
and calculates intersections for all flights in a FlightSet regardless of the source. Parameters exist to control the performance
and accuracy of the results as indicated by Table 1 and detailed in Sect. S2.2.3 of the ESM.

Results are stored in the fields of Intersection. Each field *intersection*, *observations*, and *accuracy* contains a DataFrame with
the spatial and temporal coordinates of the intersection, measured data in the vicinity of the intersection, and indicators for the
accuracy of the calculation, respectively. The DataFrame columns consist of different parameters in each category, DataFrame
rows hold data for different intersections. The different fields are linked through an ID, which makes identification of data
belonging to the same intersection easier than solely having to rely on identification by DataFrame row. Table 2 explains the
output format. An additional field with metadata exists in struct Intersection detailing the conditions of the TrackMatcher run
(see also Fig. S4 in the ESM).

### 2.5.4 Adapting TrackMatcher

The TrackMatcher package works as is for the track data described in this article with the prior installation of a licensed
MATLAB version and an installation of the PCHIP.jl package (https://github.com/LIM-AeroCloud/PCHIP.jl.git). If MATLAB
is installed in the default system folder, the link to Julia should work without further setup. Otherwise, users need to turn to the
installation guide of the Julia MATLAB package (https://github.com/JuliaInterop/MATLAB.jl.git).

For a correct data processing, data formats and file naming conventions need to be adhered. Particularly, folder and file
names of satellite data need to include keywords *CPro* or *CLay* and, if both data types are saved, need to be identical except
for those keywords (see also Note 4 in the ESM). Aircraft track data from web content need to include the flight ID, start date
of the flight, and the ICAO codes for the origin and destination (see Note 2 in the ESM).





**Table 2.** Column names in the DataFrames of the Intersection fields together with the corresponding data types and units.

| Name | Unit | Data type | Meaning |
|---|---|---|---|
| **All fields** | | | |
| id | — | String | identification for each intersection |
| **Field *intersection*** | | | |
| lat | ° | <: AbstractFloat | latitude |
| lon | ° | <: AbstractFloat | longitude |
| alt | m | <: AbstractFloat | altitude |
| tdiff | [1] | Dates.CompoundPeriod | delay between overpass times of primary/secondary trajectory |
| tprim | [2] | DateTime | time of primary trajectory at intersection |
| tsec | [2] | DateTime | time of secondary trajectory at intersection |
| atmos_state | — | Union{Missing,Symbol} | Meteorological conditions at intersection (considering altitude) |
| **Field *tracked*** | | | |
| primary | — | <: PrimaryTrack | Measured/tracked flight data near the intersection |
| CPro | — | CPro | Measured cloud profile data near the intersection |
| CLay | — | CLay | Measured cloud layer data near the intersection |
| **Field *accuracy*** | | | |
| intersection | m | <: AbstractFloat | Indicater for accuracy of the intersection calculation[3] |
| primdist | m | <: AbstractFloat | distance between intersection and nearest track point of primary trajectory |
| secdist | m | <: AbstractFloat | distance between intersection and nearest track point of secondary trajectory |
| primtime | [1] | Dates.CompoundPeriod | time difference between time of track points at intersection/nearest measured point regarding primary dataset |
| sectime | [1] | Dates.CompoundPeriod | time difference between time of track points at intersection/nearest measured point regarding secondary dataset |

[1] Units are given in the CompoundPeriod and range from ms to years; [2] DateTime in the format "yyyy-mm-ddTHH:MM:SS"; [3] Accuracy level is derived by using the interpolated trajectory of the primary and secondary dataset to calculate the complete coordinates of the intersection. The difference in meters between both calculations is the accuracy parameter.

Users outside Central Europe will have to add time zone support in the main file TrackMatcher.jl as explained in Sect. 2.2.1 of the ESM with details given in Note  andcode example 1. This allows users to operate with a *ZonedDataTime* for web data saved as local time.

In general, structs within the type tree presented in Fig. 1 exist to load and store input data from the primary and secondary trajectories or observations near intersections. Further structs calculate and store intersection data and meta information.

When adapting code, developers should obey this general structure. If the file format of a database has changed, respective routines should be updated. To add new track data, new track structs should be added to an existing or newly established





PrimarySet or SecondarySet. Section S3 in the ESM helps developers to understand the general structure of source files and the contained functions.

Data should be added in a way that all track data of the same data set are stored in a unified format. For example, all track data of the FlightSet are stored as a FlightData struct regardless of the source of the flight data. If users want to add data from another source, these data should be stored in FlightData structs as well saving identical information. If the need arises to save additional data, structs responsible for storing the data should be altered. If the data are not available in previous databases, filler objects such as *missing*, *nothing* or *NaN* should be used.

## 330   3   Application and evaluation of the TrackMatcher package

This section presents three example applications of the TrackMatcher algorithm related to the authors' research focus of finding intercepts between the ground-track of the CALIPSO satellite and (i) tracks of individual aircraft from the regional data set used by Tesche et al. (2016), (ii) one month of aircraft tracks from a global flight inventory for the year 2012, and (iii) tracks of individual clouds as identified from geostationary observations (Seelig et al., 2021). The first two applications are focussed on

studying the effect of aviation on climate (Lee et al., 2021) while the third application marks a novel approach for investigating aerosol-cloud interactions (Quaas et al., 2020), particularly the effect of aerosols on the development and lifetime of clouds.

### 3.1   Revisiting intercepts determined by Tesche et al. (2016)

Tesche et al. (2016) studied the effect of contrails that formed in already existing cirrus clouds. For their work, they investigated 37799 flight tracks for three round-trip connections from airports in California, USA (Los Angeles, San Francisco, and Seattle)

to Honolulu, Hawaii, USA in the years 2010 and 2011. Intercept points were calculated for these tracks with the CALIPSO satellite ground track. While Tesche et al. (2016) used a method for finding intercepts between the two tracks that was fit for purpose, this method was less sophisticated and less generalised than is now realised in TrackMatcher.

    Figure 3 shows a gridded map of the occurrence rate of intercept points in the data set used by Tesche et al. (2016) as identified with TrackMatcher for a time difference of $\pm 2.5\,\mathrm{h}$ together with the absolute difference compared to the intercepts

found by Tesche et al. (2016). Note that these intercepts refer to all-sky conditions and don't require cirrus to be present at flight altitude.

    Tesche et al. (2016) identified most intercepts close to the airports where the density of aircraft position data is highest. Their coverage is much sparser over the ocean where aircraft locations are often provided only every hour along the geodetic flight track (not shown). TrackMatcher's improved spatial interpolation of the flight tracks at cruising altitude compared to the

approach by Tesche et al. (2016) leads to a considerable increase in the number of identified intercept points over the ocean. Figure 3b reveals a decreased number of intercepts in the vicinity of Hawaii while one would expect an increased intercept count throughout the covered area. It is therefore likely that the number of intercepts within that small region is overestimated in the data set by Tesche et al. (2016).



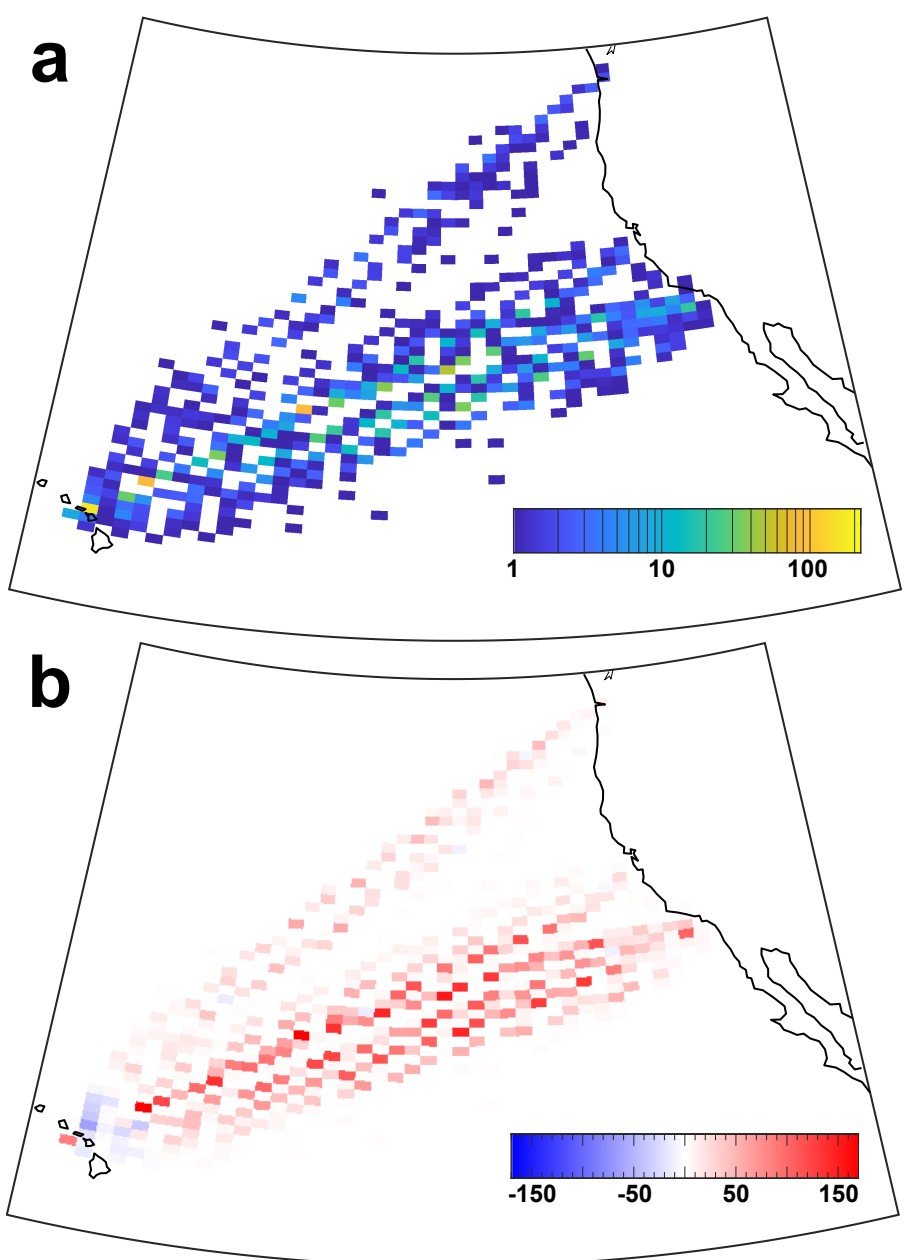

**Figure 3.** Occurrence rate of (a) intercept points between aircraft flight tracks from Honolulu to Los Angeles, San Francisco and Seattle for the years 2010 and 2011 and the ground track of the CALIPSO satellite for all-sky conditions as identified using TrackMatcher for a time difference of $\pm 2.5\,\text{h}$ and (b) absolute difference in the occurrence rate of intercepts in (a) compared to those used by Tesche et al. (2016). Red colour marks an increase in the number of identified intercepts using TrackMatcher compared to the data set of Tesche et al. (2016).





**Table 3.** Number of intercepts between aircraft tracks and the CALIPSO satellite track for the connections by Tesche et al. (2016) for different time delays and cirrus presence at flight level as identified in their and this study. Note that Tesche et al. (2016) did not investigate embedded contrails for time delays larger than $\pm 0.5$ h.

| Time delay | cirrus | Tesche et al. (2016) | TrackMatcher |
|------------|--------|----------------------|--------------|
| $\pm 0.5$ h | no | 678 | 3533 |
| $\pm 0.5$ h | yes | 122 | 291 |
| $\pm 2.5$ h | no | 3331 | 14929 |
| $\pm 2.5$ h | yes | — | 1190 |

Table 3 summarizes the statistics of applying TrackMatcher to the data set of Tesche et al. (2016). In the original study, 678
and 3331 intercepts were found for time differences of $\pm 0.5$ h and $\pm 2.5$ h, respectively. In addition, cirrus clouds had to be observed in the height-resolved CALIPSO lidar data at the altitude of a passing aircraft along the satellite track in the vicinity of the intercept point. The effects of aircraft on the properties of already existing cirrus clouds in a region of low air-traffic density were only investigated for a time difference of $\pm 0.5$ h for which a total of 122 matches could be found. Applying TrackMatcher to the same data set gives 3533 and 14929 intercepts for time differences of $\pm 0.5$ h and $\pm 2.5$ h between aircraft and satellite
passage, respectively. The constraint of having cirrus clouds at flight level to infer information of embedded contrails reduces the number of matches to 291 and 1190, respectively. This means that the use of TrackMatcher leads to an increase of suitable data of a factor of about 2.5 compared to Tesche et al. (2016).

### 3.2 One month of global flight inventory data

Next, TrackMatcher is applied to waypoint data of all civil aircraft during February 2012 from a global set provided by the US
Department of Transportation (DOT) Volpe Center. This data set was produced in support of the objectives of the International Civil Aviation Organization (ICAO) Committee on Aviation Environmental Protection $CO_2$ Task Group. It is based on data provided by the US Federal Aviation Authority (FAA) and EUROCONTROL and was used, for instance, by Duda et al. (2019). The waypoint data inventory includes time, latitude, longitude, and altitude of individual commercial aircraft for the year 2012. Volpe Center has compiled similar global data sets for the years 2006 and 2010 (Brasseur et al., 2016).

Using a global chorded data set allows to apply the approach of Tesche et al. (2016) for linking effects of embedded contrails to individual aircraft to both regions of low and high air-traffic density. The occurrence rate of intercept points identified from using the global flight inventory for February 2012 as primary and the CALIPSO satellite ground track of the same month as secondary data input for TrackMatcher are shown in Fig. 4. TrackMatcher identifies a total of 77635 intercepts between the CALIPSO ground track and the tracks of civil aircraft for a time delay of $\pm 30$ minutes (Fig. 4a).

Naturally, the largest number of intercepts are found where flight density and contrail coverage are highest (see, e.g., Fig. 1 in Duda et al., 2019). Nevertheless, there is also a considerable amount of intercepts in regions of lower air-traffic density. In particular, distinct connections, such as between Hawaii and the continents or between Australia and New Zealand, can clearly




be identified in Fig. 4a. The number of cases is reduced to 5076 in Fig. 4b as this data set includes the demand for the detection of a cirrus clouds at flight level at each intercept. This decreases the number of identified intercepts by a factor of about 15.
Using a global way point database is likely to yield a data set that is large enough to introduce sub-categories into the statistical analysis of the effect of embedded contrails on cirrus clouds.

The findings in Fig. 4 demonstrate that TrackMatcher can be applied to data sets of considerable size. However, parallel computation is not yet achieved in TrackMatcher with the exception of file reading with the CSV package. To achieve appreciable performance, the test month was split into four segments each of about a week's length. Accumulated file loading times
are approximately 50 minutes for the flight data and 2 minutes and 26 seconds for the satellite data. Overall, 2.25 million flights were loaded and processed to be saved in a unified format from 121 GB of data. 3.8 million satellite data points were extracted

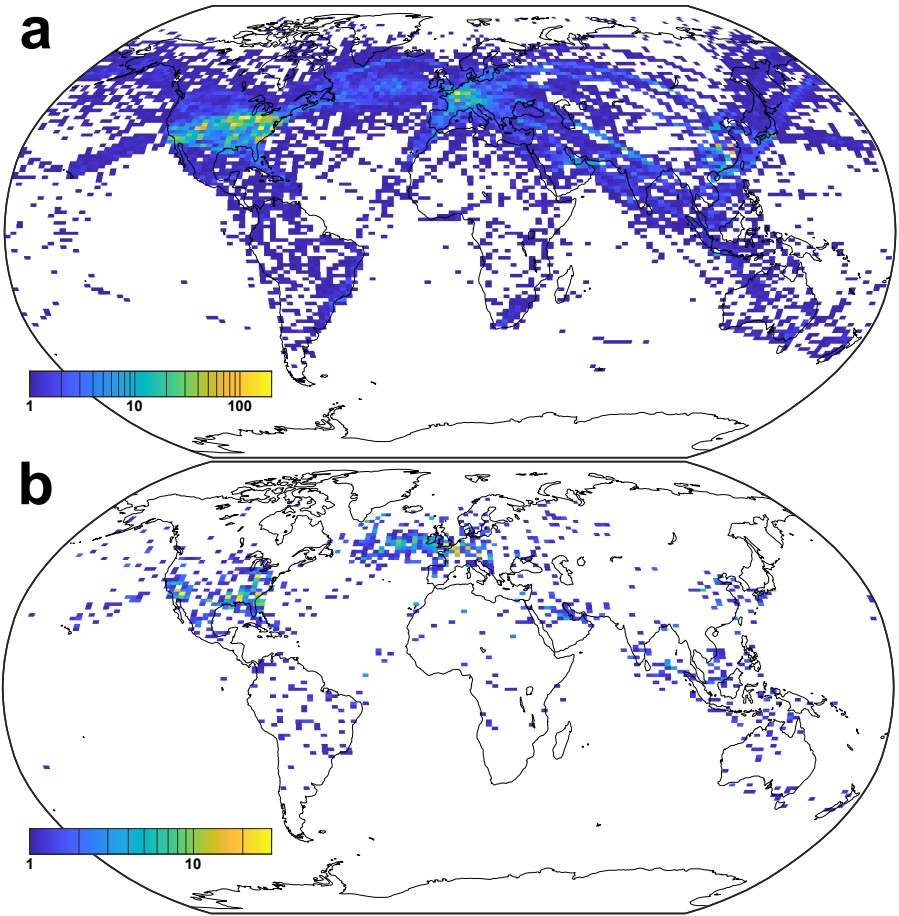

**Figure 4.** Gridded map (0.5° latitude by 1.0° longitude) of intercept points between the aircraft flight tracks above 5 km height and the ground-track of the CALIPSO satellite for February 2012 for (a) all atmospheric conditions and (b) situations in which cirrus is present at flight level. The maximum time difference between the primary and secondary tracks is constrained to 30 min.



and processed from 89 GB of CALIOP cloud profile data. The combined processing time to compute intersections was 3 days, 9 hours, and 22 minutes.

Such a large data set allows for a detailed analysis of the precision of TrackMatcher calculations. TrackMatcher stores several accuracy parameters to evaluate the results. Parameters include the time and spatial distance to the nearest observed primary and secondary track point, and an indicator for the precision of the calculation. This indicator is not the result of an error propagation. Instead, TrackMatcher takes the calculated $x_0$ value for the determined root in the distance function (Eq. (1)) and calculates the corresponding $y_0$ from the interpolated trajectories of the primary and secondary tracks. Using the haversine function (Eq. (12)), TrackMatcher calculates the distance between both computations of the intercept points using either the primary or secondary data set. The distance between both computations is the indicator for the accuracy of the calculation. This indicator only recognises the accuracy of the interpolation method and the calculation of the roots in the distance function. It does not consider any measurement errors in the track data.

Table 4 shows minimum, maximum, mean, and important quantiles for the accuracy indicator and the spatial and temporal distance to the nearest observed track point in the results of the TrackMatcher run for February 2012. Generally, CALIPSO satellite data of the secondary data set have a much greater density of track points, which is resembled by a median distance of about 1 km to the nearest measured track point compared to the median distance of 11 km for the primary flight data. Both data sets show some large data gaps resulting in a maximum distance of 2615 km and 4648 km of the computed inter section to the nearest observed track point of the primary and secondary data set, respectively. Due to the much greater velocity of the satellite compared to an aircraft, the maximum time difference to the nearest measured track point is only about 8 minutes for the secondary data compared to 6 hours and 50 minutes for the primary data. Larger data gaps are more common in the primary data, which is represented by the fact of the mean in the range of the 68th percentile. For the secondary data, the mean corresponds to the 0.986-quantile. Hence, only about 1% of the track points is above the average of 1 km and with a

**Table 4.** Statistics on the accuracy indicator as well as the spatial and temporal distances of the calculated intersection to the nearest measured track point of the primary or secondary data set for the run using data of February 2012 by the DOT Volpe Center. The units for the accuracy indicator and distances are meter.

|  | Accuracy indicator of intercept calculation | Distance to nearest track point | | Time difference to nearest track point | |
|---|---|---|---|---|---|
|  |  | Primary track | Secondary track | Primary track | Secondary track |
| minimum | 0.0 | 0.0 | 0.0 | 0 s | 0 s |
| lower quartile | 0.0 | 3665 | 466.89 | 15 s | 92 ms |
| mean | 47.9 | $2.83 \cdot 10^4$ | 7491.6 | 1 min, 38 s | 1 s |
| median | 0.28 | $1.10 \cdot 10^4$ | 1034.1 | 41 s | 188 ms |
| upper quartile | 0.79 | $3.78 \cdot 10^4$ | 1949.1 | 2 min, 16 s | 282 ms |
| maximum | $5.48 \cdot 10^5$ | $2.62 \cdot 10^6$ | $4.65 \cdot 10^6$ | 6 h, 50 min | 8 min, 11 s |





maximum distance of 4648 km dominantly impacting the mean, most distances between the calculated intersection and the nearest measured track point are significantly smaller.

TrackMatcher performs remarkably well with most results being calculated with an accuracy of only a few meters. The median accuracy is 0.28 m, and the mean accuracy of 48 m is in the range of the 98th percentile. 243 out of the 77635 intersections or 0.3% are computed with an accuracy larger than 1000 m, 1869 or 2.4% are above 10 m, and 13352 or 17.2% are above 1 m of accuracy.

### 3.3 One year of cloud tracks over the Mediterranean

TrackMatcher is also useful for supporting aerosol-cloud interaction studies. Specifically, tracks of individual clouds as inferred from time-resolved observations with an instrument aboard a geostationary satellite should be matched with the tracks of polar-orbiting satellites that provide a highly detailed snapshot observation of the same clouds once during their life time. Here, cloud tracks in the region spanning from 16.5°W, 28.7°N to 34.3°E, 59.8°N during January to December of 2015 as identified in the CM SAF CLoud property dAtAset using SEVIRI (CLAAS-2) data set (Benas et al., 2017) following Seelig et al. (2021)

have been used as primary input into TrackMatcher. The tracked clouds in this data set (i) are low-level clouds that (ii) formed in clear air and (iii) dissolved in clear air. All these clouds could be followed throughout the entirety of their lifetime. Clouds that originate from splitting or end as merged clouds are not tracked with the current methodology. Figure 5 shows the cloud tracks together with the identified intersections.

In contrast to aircraft trajectories, the success rate of finding intersections in cloud tracks is significantly reduced. The main

reason is, that TrackMatcher compares sets of trajectories. Therefore, the centre points of a cloud trajectory have been matched to the satellite ground track. For clouds with a large horizontal extent, this means a high chance of the satellite monitoring the cloud, but not being identified by TrackMatcher, when the satellite does not pass the centre line of the cloud. A new method needs to be developed to compare an area or volume to a trajectory to increase the efficiency of the success rate in TrackMatcher. Another significant reason for the reduced number of matches are much shorter trajectories. With average cloud

life times of about 2 hours (Pruppacher and Jaenicke, 1995) and with track points every 15 minutes, about 8 track points per cloud can be expected. With the above mentioned pre-conditions further reducing the primary data, median cloud trajectory length is 4 track points. Overall, the data set consisted of over 1.7 million trajectories.

To increase chances for finding intersections, the TrackMatcher run for cloud tracks allowed a maximum delay time of 5 hours at the intersection between the overpass times of the primary and secondary trajectory. TrackMatcher was able to identify

2969 intersections. 1527 intersections were within the default delay time of $\pm0.5$ h. This corresponds to a success rate of 0.8‰ and 1.7‰ for standard and extended conditions, respectively.

### 3.4 Sensitivity study of key parameters

As indicated in in Sect. 2.5 and by Table 1, parameters exist to compromise between the algorithm's performance and the accuracy of results. To investigate the influence of different settings, several sensitivity studies have been performed, where

one parameter was varied from standard conditions. Table 5 summarises computation time and identified intersections for the

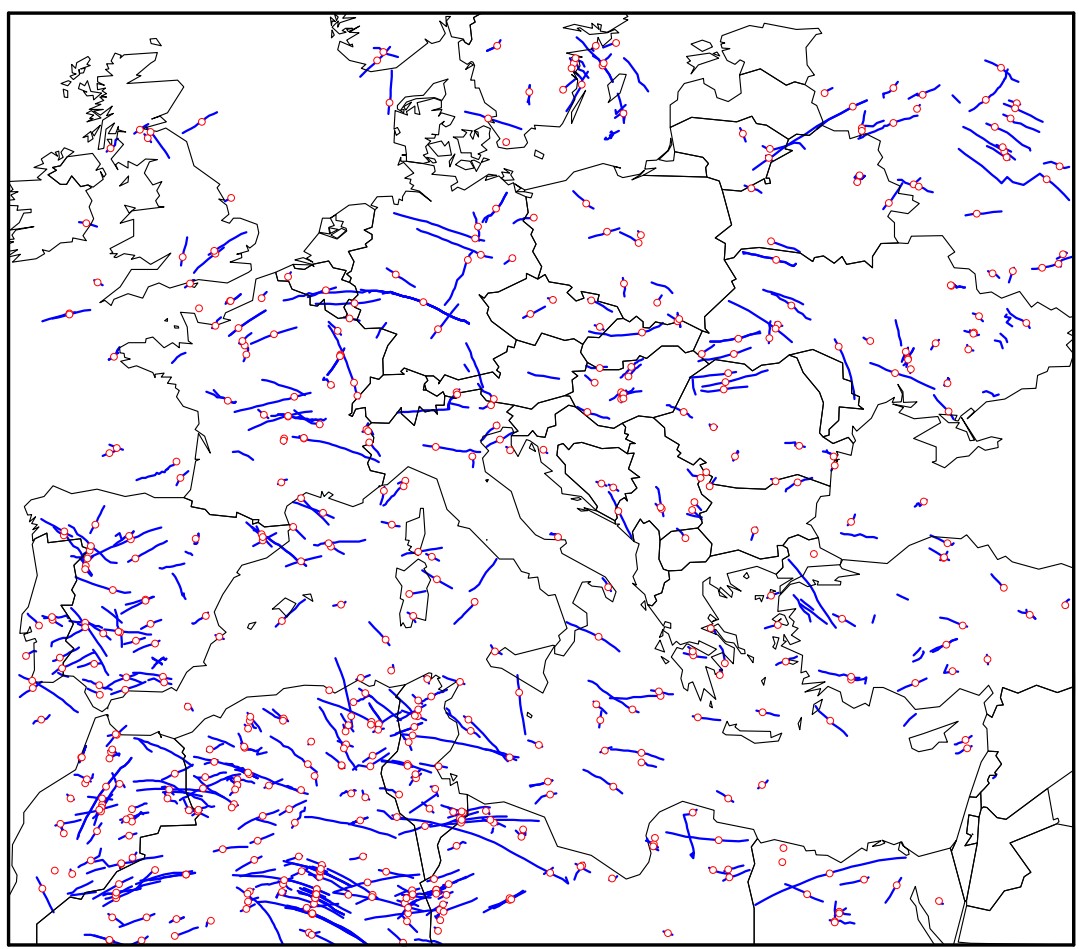

**Figure 5.** Intercept points (red circles) between all cloud tracks (blue lines) and the CALIPSO ground track (not shown) identified in the region spanning from $16.5°$W, $28.7°$N to $34.3°$E, $59.8°$N during 2015.

various studies. All sensitivity runs used flight track data from 1. January 2012 of the DOT Volpe data set containing almost 67000 flight tracks. CALIOP profile data was used for the same day, except for the 2 runs investigating the use of CALIOP layer data.

Under default conditions, TrackMatcher finds 2485 intersections in 2 hours and 33 minutes. 295 intersections include con-
ditions other than clear sky or a missing signal. Most of the sensitivity runs using cloud profile data show a similar run time and similar success rates. Exceptions are the run with an increased maximum delay time between the primary and secondary trajectory overpass at the intersection finding 3.7 times more intersections and the run with a finer resolved interpolation step width. Both cases see a massive increase in computation time to 6 hours and 47 minutes and 6 hours and 21 minutes for varying the maxtimediff and stepwidth parameter, respectively. The increase in computation time for increasing the maximum
time difference can be explained by an increase of intercept finds to 9239. Surprisingly, the finer resolved interpolation step

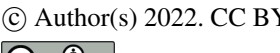



**Table 5.** Run times and number of identified intersections for various sensitivity studies. The first column lists parameter settings; Float64 means all input data is loaded in double precision, CLay means satellite data is loaded as layer data rather than profile data with the additional condition of loading input in double precision in scenario CLay64. Parameters are applied during the calculation of intersections unless otherwise indicated.

| Scenario | run time | | Intersections determined by TrackMatcher | |
|---|---|---|---|---|
| | | total | with atmospheric features | no signal/missing values[1] |
| default | 2 h, 33 min | 2485 | 295 | 18 |
| Float64[2] | 2 h, 25 min | 2487 | 296 | 19 |
| CLay[2] | 1 h, 3 min | 2485 | 282 | 2203 |
| CLay64[2] | 1 h, 4 min | 2487 | 283 | 2204 |
| maxtimediff = 150 | 6 h, 47 min | 9239 | 1028 | 48 |
| altmin = 500[2] | 2 h, 34 min | 2737 | 305 | 18 |
| expdist = 100000 | 2 h, 34 min | 2335 | 279 | 18 |
| Xradius = 0 | 2 h, 34 min | 2494 | 295 | 18 |
| stepwidth = 0.1 | 2 h, 25 min | 2396 | 284 | 17 |
| stepwidth = 0.001 | 6 h, 21 min | 2350 | 275 | 17 |
| atol = 1 | 2 h, 33 min | 2508 | 302 | 18 |
| atol = 0.01 | 2 h, 34 min | 2437 | 293 | 17 |

[1]Cloud layer data only recognises atmospheric features and does not distinguish between clear sky conditions and no signal. Any non-featured values are treated as missing in TrackMatcher. [2]Parameter is applied during loading of input data.

width results in less identified intersections (2350). In essence, handing too many and too finely resolved data points to the IntervalRootFinding package results in a performance loss and sometimes failure to identify intersections.

Using cloud layer data instead of cloud profile data decreases computation times by a factor of 2.5. The same number of intersections are found, however, meteorological conditions are not always correctly identified (282 finds under non-clear conditions compared to 295 using cloud profile data). Moreover, it is not possible to distinguish data without meteorological features using layer data. Cloud profile data holds additional information such as clear sky conditions or no lidar signal. On the other hand, using cloud layer data, results in significantly reduced file sizes of the stored output (63.8 MB for layer data compared to 992.5 MB for profile data). In rare cases, the floating point precision of the input data can have an effect on the results, and two additional intersections are found by TrackMatcher when using double precision. While computation times are not affected by the floating point precision, switching to double precision increases the size of the output files with negligible effects on the quality of the results. This effect is only seen for cloud profile data, where file sizes of saved output increased from 992.5 MB to 1.47 GB. File sizes for cloud layer data were almost identical (63.8 MB compared to 66.2 MB).

The number of false duplicate intersection identification can be inferred from the TrackMatcher run setting the Xradius parameter to zero. Intersections increase to 2494, hence, nine duplicate intersections are falsely determined. Currently, TrackMatcher saves all intersections regardless of the distance to the nearest track point. Data can be filtered in a post analysis as the





distance to the nearest track point is saved in the accuracy field of the Intersection struct. Furthermore, data could be filtered by the accuracy of the computation. Limiting the maximum distance to the nearest observed track point to 100 km decreases the determined intersections to 2335. The maximum accuracy indicator of 5598,8 m is identical to the base scenario; median accuracy of 0.28 m and mean accuracy of about 8 m are similar. This leads to the conclusion that the PCHIP method is very
accurate even with large gaps in the data. Other factors influencing the precision of the results could be sharp bends in the trajectories, or inaccuracies in the track data leading to discontinuities.

## 4   Conclusions and outlook

This paper present a tool for finding intercept points between tracks of geographical coordinates, i.e. data sets consisting of at least time, latitude, and longitude. The main principles of the methodology consist of (i) interpolating the primary and
secondary tracks with a piecewise cubic Hermite interpolating polynomial and (ii) finding the minimum norm between the different track point coordinate pairs. The universal design of TrackMatcher allows application of the tool to a wide range of scenarios such as matching tracks from ships, aircrafts, clouds, satellites, and in fact any other moving object with known track data.

Here, the tool is applied to find intercept point between flight tracks of individual aircraft and satellite ground tracks as
well as between tracks of individual clouds and satellite ground tracks. Potential application of the TrackMatcher tool in atmospheric science include the identification of intercept points between the tracks of research aircraft and the tracks of clouds or trajectories of air parcels from dispersion modelling. TrackMatcher will also prove useful in research fields outside the atmospheric sciences whenever data collected along different spational pathways need to be collated with an objective and reproducible methodology.

However, current studies have also shown limitations of TrackMatcher. Identification of intercept points matching cloud data with other trajectories can be improved with the development of an algorithm comparing trajectories and areas or volumes. Further development will also focus on performance improvements, e.g., by enabling distributed runs or parallel computation.

*Code availability.*   The TrackMatcher package is available at https://github.com/LIM-AeroCloud/TrackMatcherPaper.git under the GNU general public licence version 3.0 or higher. The TrackMatcher code for release v0.5.3 including raw data and plotting scripts for the result
figures can be obtained from Zenodo https://doi.org/10.5281/zenodo.6193048. Routines concerning the interpolation of track data with the PCHIP method are available in a separate package at https://github.com/LIM-AeroCloud/PCHIP.jl.git under the same licence. Release v0.2.1 is available at https://doi.org/10.5281/zenodo.6193059.

*Author contributions.*   TrackMatcher is based on an idea by MT and PB. The code development is lead by PB. All authors contributed equally in the processing and interpretation of the data as well as in the preparation of the manuscript.



*Competing interests.* The authors declare no competing interests.

*Acknowledgements.* This work was supported by the Franco-German Fellowship Programme on Climate, Energy, and Earth System Research (Make Our Planet Great Again – German Research Initiative, MOPGA-GRI) of the German Academic Exchange Service (DAAD), funded by the German Ministry of Education and Research. The authors thank Torsten Seelig and Felix Müller for providing the cloud trajectories used in the third application example. We thank Gregg G. Fleming of Volpe Center for providing the aircraft waypoint data set for 2012.





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
