# Peer review of "TrackMatcher — A tool for finding intercepts in tracks of geographical positions"

_Geoscientific Model Development, 2021_

## Referee Comment (RC1)

**Review of "TrackMatcher" GMD manuscript by Bräuer & Tesche**

Summary:

The manuscript presents a tool to find intersects of data points in two types of data sets (as exemplified by aircraft or cloud tracks and satellite data with narrow swath).

The manuscript is well-written and concise. The problem that is solved and the algorithm itself are not overly complex, but I believe still worth to be published in such a separate publication as this give room for sensitivity studies that are do usually not appear in other journals.
I have only some minor concerns and recommend accepting the manuscript once those are resolved.

Minor issues:

1. Some sections I regard as too technical, even for GMD standards. The dependencies in section 2.2 could be removed from the paper and only listed on the GitHub/Zenodo README file. Furthermore, section 2.3. is very technical, at least for someone who is not familiar with Julia. It is good to define what is meant with all those technical terms that are written in italics. But I question a bit the necessity to talk about constructors at that stage. If a reader finds this information helpful clearly depends on whether he/she wants to understand how your algorithm works or intends to use or extend the existing code.
2. How/where is the term „continuous" used in line 42 defined?
3. Around line 200: It is not clear to me, how the width and height of the bounding boxes are computed.
4. Fig. 2: In your example, 4 dots are yellow (acceptable time diff) and one is red (delay to large). Is this typical that spatial intercepts have an acceptable delay? I would imagine, that in most cases, the time difference of the collocated observations is too large. Or this example a special case where aircraft needs about the same time as the satellite to jump from one satellite track to the next?
5. line 350: To me it does not sound reasonable that using a different interpolation scheme increases the number of intercepts. I understand that the coordinates of interceptions may change. Moreover, it is conceivable that intercepts are only identified for one of the two interpolation schemes. But I do not see why Hermitian interpolation give such a strong bias towards more intercepts. Could it be that there was bug in the previous algorithm or some other parameters may have changed between the two version you compare? What happens, when TrackMatcher alternatively uses a linear interpolation scheme?
   With the new intercept data, do the findings of Tesche et al, 2016 still hold? Can you give at least a short summary on this in this manuscript?
6. line 451-452: ambiguity of the term performance: do you mean run time or the algorithm success? Apart from this wording issue, I do not understand, why your algorithm does not work well for a finer resolution.
7. comment to line 460: One could also do the computations in double precision and the output in single precision.

Technical issues:

1. The abbreviation ESM is nowhere defined.
2. line 146: Eq 1??
3. line 169: "source code" or "source code execution"?
4. last line of page 7: acceptable

5. line 216: replace "set" by "specified"
6. line 385: it would fair to write "2.5 min"
7. Caption of Fig.4: Gridded map of THE NUMBER of intercept points?
8. line 479: pointS
9. line 483: spational -> spatial?

---

## Author Comment (AC1)

We thank the two reviewers for the time and effort in reviewing our manuscript and for providing very positive reviews. The valuable comments were incorporated into the paper and helped to improve the quality of the publication. Below are our point-by-point replies (blue) to the comments (black).

**Anonymous Referee #1**

Minor issues:

1. Some sections I regard as too technical, even for GMD standards. The dependencies in section 2.2 could be removed from the paper and only listed on the GitHub/Zenodo README file. Furthermore, section 2.3. is very technical, at least for someone who is not familiar with Julia. It is good to define what is meant with all those technical terms that are written in italics. But I question a bit the necessity to talk about constructors at that stage. If a reader finds this information helpful clearly depends on whether he/she wants to understand how your algorithm works or intends to use or extend the existing code.

We have simplified section 2 to address the first comment:

- The list of package dependencies has been removed from section 2.2 and is already covered in TrackMatcher's WIKI.
- We thank Reviewer # 1 for pointing out the need for clarifications for non-Julia users. We have simplified the language in Section 2.3 and added additional clarifications with most significant changes in the second paragraph of Section 2.3. However, we believe that most of the information in this section is necessary and crucial to understand the workflow of TrackMatcher. The intention of this publication is to deliver a starting point for applying and potentially adopting TrackMatcher. Constructors play a crucial role in TrackMatcher as they initiate each process such as data import or computation of intercept points. Moreover, a minimum level of understanding of Julia is required to operate TrackMatcher. However, we hope that the simplified language and extra explanations help to convince non-Julia users to try out this tool and acquire a minimum knowledge of the Julia language.

2. How/where is the term „continuous" used in line 42 defined?

We wanted to emphasize the difference between primary data that consists of several individual (usually short or at least shorter) tracks and secondary data that only holds data of a single potentially very long trajectory. We have rephrased the term continuous with "*single and potentially very long*" to avoid confusion.

3. Around line 200: It is not clear to me, how the width and height of the bounding boxes are computed.

The bounding box is derived from the minimum and maximum lat/lon values of the track data. Explanations about this derivation have been added to ll. 194–199 in the revised manuscript.

4. Fig. 2: In your example, 4 dots are yellow (acceptable time diff) and one is red (delay to large). Is this typical that spatial intercepts have an acceptable delay? I would imagine, that in most cases, the time difference of the collocated observations is too large. Or this example a special case where aircraft needs about the same time as the satellite to jump from one satellite track to the next?

The CALIPSO satellite has a period of 98.5 min and a mean motion of 14.57°. The flight is in the northern mid-latitudes roughly between 45°N and 60°N, where a 1° distance between meridians decreases from about 78 km to about 55 km. To follow the satellite, an aircraft would have to fly at a velocity of roughly 55-80 km * 14.57 / 1.5 h or 534 - 777 km/h, which is in the range of typical aircraft

ground speeds. The additional allowed time difference of 30 min gives an extra buffer. So, planes flying nearly parallel to circles of latitudes and following the satellite can be matched several times to CALIPSO observations. However, the plot is intended to demonstrate the principle functions of TrackMatcher. We have thus refrained from giving this detailed explanation in the article as it would not add any value within the scope of the publication.

5. line 350: To me it does not sound reasonable that using a different interpolation scheme increases the number of intercepts. I understand that the coordinates of interceptions may change. Moreover, it is conceivable that intercepts are only identified for one of the two interpolation schemes. But I do not see why Hermitian interpolation give such a strong bias towards more intercepts. Could it be that there was bug in the previous algorithm or some other parameters may have changed between the two version you compare? What happens, when TrackMatcher alternatively uses a linear interpolation scheme?

With the new intercept data, do the findings of Tesche et al, 2016 still hold? Can you give at least a short summary on this in this manuscript?

The problem with aircraft position data over the ocean is that they are not measured with Automatic Dependent Surveillance—Broadcast (ADS-B) receivers but generally interpolated geodetically with a distance of about one hour between consecutive points. In Tesche et al. (2016), the matching was performed by reducing the distance between individual points of the aircraft trajectory and the satellite track. In matching points, a finer spacing in the geodetic interpolation of the unevenly spaced aircraft trajectory would produce a larger and closer spaced number of points to investigate for their distance to the evenly spaced points (5 km distance) of the CALIPSO satellite track. This is now realised in TrackMatcher.

Tesche et al. (2016) have applied additional filters to make sure that only otherwise homogeneous cirrus clouds are investigated for the effect of penetrating aircraft. Such an elaborate screening has not been performed in the evaluation of the performance of TrackMatcher presented in our paper. Nevertheless, we are confident that the findings of Tesche et al. (2016) still hold. To elaborate on this issue, we have revised the text in Section 3.1.

6. line 451-452: ambiguity of the term performance: do you mean run time or the algorithm success? Apart from this wording issue, I do not understand, why your algorithm does not work well for a finer resolution.

We have reworded "performance loss and sometimes failure to identify intersections" to "performance loss in terms of run time and sometimes even algorithm success" to reduce ambiguity.

An explanation for this behaviour is not immediately available and needs further investigation. It could be the result of the chosen PCHIP method for track interpolation. PCHIP tries to interpolate between track points by drawing a graph that is forced through every track point, which is then given to the IntervalRootFinding package to find an intercept points with the satellite track. Some flight tracks have highly inaccurate track points and overfitting cannot be avoided with the current method. Therefore, a finer resolution could lead to slightly different trajectory that does not cross the satellite ground path any more or is outside the allowed time interval. Currently, TrackMatcher looks for intercepts by finding roots, i.e. where the distance equation (1) in the paper becomes exactly zero. Allowing some margin of error in this calculation in future versions of TrackMatcher might overcome this issue.

7. comment to line 460: One could also do the computations in double precision and the output in single precision.

This is a valid point. Currently, some calculations of finding the intercepts are forced to double precision, even if single precision is chosen to ensure accurate calculations. The wiki of TrackMatcher will be adjusted to ensure the communication to the end user. Using single precision is a compromise to the above suggestion. For the most important calculations, double precision is ensured, but it is derived from single precision input. Usually, disk space is not so much of an issue for TrackMatcher output, and conversion within the programme to single precision would result in increase computation times. Therefore, data could be post-processed to reduce file size without affecting computation times of the actual model run.

Technical issues:

1. The abbreviation ESM is nowhere defined.

The unabbreviated form together with the abbreviation in parentheses has been given at the first use as 'electronic supplementary material (ESM)' in line 82 of the revised manuscript.

2. line 146: Eq 1??

The mislabelled latex reference has been corrected to point to Eq. 1.

3. line 169: "source code" or "source code execution"?

Actually, both is true. But in this context, we meant the large number of code lines. For more clarity, this portion of the code was moved to a separate package. As we only want to focus on this aspect, we kept the original phrase.

4. last line of page 7: acceptable

We have corrected the typo.

5. line 216: replace "set" by "specified"

The word 'set' was replaced by 'specified'.

6. line 385: it would fair to write "2.5 min"

The phrase '2 minutes and 26 seconds' was replaced by '∼ 2.5min'.

7. Caption of Fig.4: Gridded map of THE NUMBER of intercept points?

The phrase 'the number of' has been inserted to the caption of Fig. 4.

8. line 479: pointS

The missing 's' to give the plural has been added to 'point'.

9. line 483: spational -> spatial?

We corrected the typo to 'spatial'.

**Anonymous Referee #2**

This paper describes TrackMatcher, software for computing the intersections between computer trajectories and satellite observations. The paper describes the software construction and design as well as providing a few example cases demonstrating the use of the software. The paper can be a bit technical in some areas, particularly for people not familiar with Julia, but the supplementary information is very useful in this case.

This paper covers a tool to help with a common task. Although this task is not particularly complicated, it is good to see the algorithm described in detail. It would benefit from a clearer statement of where TrackMatcher improves over previous work (or what makes it necessary). Given this is a common task, is there something that previous studies were missing?

This work is within scope for GMD and I would support publication after a few corrections.

L49 - Although the package is described in more general terms, it is clearly designed for matching calculated trajectories with CALIPSO data. It might be good to put this description of primary and secondary first (which the terms are defined). I would have found this helpful for remembering which type of data the primary and secondary values are.

We have added a description of the currently available trajectory data sets to section 2.1 on ll. 42 – 46.

L70 - It might be nice to distribute some very simple test data with the package, so that contributors can be sure that any changes they make don't break the base functionality of the package.

We have added an example with flight data in the ESM. For legal reasons and because of the file size, we could not provide satellite data, but added an explanation how to obtain satellite data free of charge. In the ESM, section 2.2.4 was added explaining the input data, how to run the example, and the expected results.

L71 - While I understand the necessity of using Matlab to read in the hdf files, it does detract slightly from the free (as in both freedom and beer) nature of the whole package. I know in python, if the netcdf library is built correctly, it can be used to read hdf4 files, too. You may not want to include this now, but if the same capability exists in julia, perhaps it is useful for a future version?

We have already initially explored several options, but have decided that this is not of such a high priority to block the publication of TrackMatcher. When time permits, it is planned to move to freely available software such as python or fortran for HDF4 file reading. To document our plans to switch to open source software before version 1.0, we have created an issue on the github repository: https://github.com/LIM-AeroCloud/TrackMatcher.jl/issues/48.

L75 - The package versions could probably be in the supplementary information.

Following the suggestion of both Referees, the list of package dependencies has been removed from Section 2.2.  It is already covered in TrackMatcher's WIKI.

L95 - Please define ESM when it is first used

The unabbreviated form together with the abbreviation in parentheses has been given at the first use as 'electronic supplementary material (ESM)' in line 82 of the revised manuscript.

L124 - Is there any reason not to use the haversine distance here? It is slightly slower, but might provide an improvement in the accuracy (especially near the poles).

The haversine function is useful to calculate distances on the surface of a sphere. It recognises the increased distance between two points on a sphere due to the curvature effect in comparison to the shortest distance going through the sphere. In Eq. (1), we want to determine the root of the distance function, where the distance between 2 points becomes zero. While we could use the haversine function, there is no need for it as there will be no effects of a curved surface, when the distance is 0. In other words, it makes no difference, whether one goes along the shortest distance on the surface of the sphere or through the sphere, if one only cares about, where the distance between two points becomes zero.

L179 - acceptable

The typo has been corrected

Figure 2 - the coloured dots at the intersections are very small on my screen.

We have increased the size of the intersection dots and forced them on top to make intersections more visible. We believe that the size of the track points is adequate as the plot focuses on the tracks rather than individual data points, but with the current point size makes it possible for at least the flight track to resolve individual track points.

L255 - 'If MATLAB is not installed at ...'?

We have rephrased to "installed in" in compliance with this stack overflow thread: https://english.stackexchange.com/questions/103542/install-on-install-in-install-to.

L361 - There are a very large number of extra matches in this study. I am not really clear how this has happened and I think this really requires some more discussion. This comparison to previous studies is the main way of assessing the accuracy of TrackMatcher, rather than the accuracy metric (which is more testing simplifications in the matching process). Was there a deficiency in Tesche et al, 2016 (or is TrackMatcher overdoing the matches here, or missing in some cases?).

This issue was also raised by the other Referee. Please see our reply to point 5 and the revised text.

L388 - Is this for the 4-core run, or a single core run of TrackMatcher?

TrackMatcher code is currently not run distributed or in parallel. Some packages imported in TrackMatcher such as CSV.jl for reading csv files use multiple cores, but effects on performance are minor and concern only the data import. All runs were performed with 2 cores, but due to the fact that we have not parallelised the code yet, we focused on short and necessary information and have refrained from mentioning the number of cores in the runs.

L404 - Given the satellite travels much faster, 8min is thousands of kilometers - is this plausible?

The period of the CALIPSO satellite is 98.5 minutes. With an Earth radius of 40,000km, 8 min would yield roughly 3250 km. This is considerably more than the 2620 km, but as stated in the paper intersection calculated within such large data gaps should be treated with extreme caution. Further, for flight track data, the more accurate PCHIP interpolation method is used, while linear interpolation is used for time. For the purpose of TrackMatcher, this limitation is acceptable as the time difference is only used for the exclusion of computed intersection and does not influence the precision of the calculation. Moreover, the focus of the study is on aircrafts in cruising altitude with a constant velocity. Therefore, the results are regarded as plausible. To limit errors, a switch exists to disregard results that are above a threshold distance from the nearest measurement.

L411 - I think this accuracy value doesn't really cover much about the accuracy of trackmatcher itself, more about the accuracy of using either lat or lon for finding an intersection. Hence this value is

more related to the shape/curve of the trajectories. Perhaps it could be referred to as something else, maybe the 'interpolation accuracy' or 'matching accuracy'?

We now use interpolation accuracy instead of accuracy.

Fig. 5 - Some of these tracks appear to have no intersections. Is that intentional?

Good catch. We have removed those trajectories.

L469 - Is this really a conclusion about PCHIP? There is no comparison to any other source of data. Perhaps removing some intermediate flight points the testing the reconstruction could test the accuracy of PCHIP (although I don't think the accuracy of PCHIP needs to be shown in much detail for this work).

Differences stem most likely from inaccuracies in the track data rather than from inaccuracies of the PCHIP method. Nevertheless, errors from PCHIP are unavoidable with the current algorithm. To better illustrate our point we have rephrased the statement to:

"Hence, inaccuracies in the computations are most likely not the result of the PCHIP method even for track data with large gaps."